# Validity of an ICD-10 Coding Algorithm for Acute Heat Illness in the Emergency Department: A Retrospective Cohort Study

**DOI:** 10.3390/ijerph21091132

**Published:** 2024-08-27

**Authors:** Hasan Baassiri, Timothy Varghese, Kristin K. Clemens, Alexandra M. Ouédraogo, Kristine Van Aarsen, Branka Vujčić, Justin W. Yan

**Affiliations:** 1Division of Emergency Medicine, Department of Medicine, Schulich School of Medicine and Dentistry, Western University, London, ON N6A 5A5, Canada; 2Lawson Health Research Institute, London Health Sciences Centre, London, ON N6C 2R5, Canada; 3Division of Endocrinology and Metabolism, Department of Medicine and Department of Epidemiology and Biostatistics, Schulich School of Medicine and Dentistry, Western University, London, ON N6A 5A5, Canada; 4ICES, London, ON N6A 5W9, Canada

**Keywords:** acute heat illness, emergency department, ICD-10 coding algorithm, retrospective cohort study

## Abstract

Acute heat illness (AHI) from extreme environmental heat exposure can lead to emergency department (ED) visits, hospitalization, and even death. While the ICD ninth revision codes for AHI have been validated in the U.S., there have been no studies on the validity of ICD-10 codes for AHI in Canada. The objective of this study was to assess the validity of an ICD-10 coding algorithm for ED encounters for AHI. We conducted a retrospective cohort study of children and adults who had ED encounters at two large academic, tertiary hospitals in London, Canada, between May and September 2014–2018. We developed an algorithm of ICD-10 codes for AHI based upon a literature review and clinical expertise. Our “gold-standard” definition of AHI was patient-reported heat exposure and documentation of at least one heat-related complaint. To establish positive predictive value (PPV), we reviewed 62 algorithm-positive records and noted which met our “gold-standard” definition. To calculate negative predictive value (NPV), sensitivity (Sn), and specificity (Sp), we randomly reviewed 964 ED records for associated ICD-10 codes and diagnoses. Two independent reviewers completed blinded data abstraction, with duplicate abstraction in 20% of the sample. Of the 62 algorithm-positive records, mean (SD) age was 38.8 (23.8) years; 37% were female. PPV was 61.3 ± 12.1% (95% CI). Of the 964 randomly selected records, mean (SD) age was 41.7 (26.5) years; 51.1% were female. The NPV was 99.7 ± 0.4%, sensitivity 25.0 ± 42.4%, and specificity 100.0 ± 0.0%. An ICD-10 coding algorithm for AHI had high specificity but was limited in sensitivity. This algorithm can be used to assemble and study cohorts of patients who have had an AHI, but may underestimate the true incidence of AHI presentations in the ED.

## 1. Introduction

The year 2020 marked the end of the warmest decade on record, with global temperatures expected to reach new records over the next five years [1,2]. Between 2000 and 2016, 125 million adults world-wide were exposed to heat waves due to an increasing frequency and severity of heat events [3].

Extreme heat exposure can have significant health consequences for adults and children. A direct health consequence of extreme heat is acute heat illness (AHI) (e.g., heat stroke, heat exhaustion, and dehydration). AHI can lead to emergency department (ED) visits, hospitalization, and even death [4,5].

To inform public health policy and planning, it is essential to understand the health impacts of climate change at a population level. Administrative codes maintained in large healthcare databases might be helpful. When patients present to a healthcare provider (e.g., have an ED visit) in Ontario, their clinical information is documented by the healthcare team in the medical chart. Trained hospital coders then assign codes to patient charts based upon documented diagnoses and procedures. Administrative codes include International Classification of Disease (ICD) 10th revision codes, which were established by the World Health Organization and have been used in Canada since 2002. In our region, we have validated administrative codes for conditions including obesity [6] and hypoglycemia [7] that have since been used reliably across several research studies [8,9,10,11]. Similar to these other medical conditions, contemporary ICD codes (i.e., ICD-10) can be a powerful tool to understand the epidemiology, outcomes, and healthcare utilization associated with AHI. Before using codes in research studies, however, it is important to understand their validity and ensure their accuracy, thereby reducing the possibility of errors and misclassification of health conditions [12].

While ICD 9th revision codes for AHI have been validated in U.S.-based studies [13,14,15], there have been no studies on the validity of ICD-10 codes for AHI in Canada or elsewhere in the world. In this study, we aimed to assess the validity of an ICD-10 coding algorithm for ED encounters for AHI in Ontario, Canada.

## 2. Materials and Methods

This study was conducted and reported in accordance with the Standards for Reporting Diagnostic accuracy studies (STARD) guidelines [16]. As this was a retrospective study of pre-existing data, the need to obtain informed consent was waived by Western University’s Health Sciences Research Ethics Board, who also approved the study protocol.

### 2.1. Study Design and Setting

We conducted a retrospective cohort study at two large academic, tertiary hospitals at London Health Sciences Centre (LHSC) in London, Ontario, Canada [17] between May and September 2014–2018. LHSC’s EDs see approximately 150,000 patients per year and have a catchment that includes Southwestern Ontario and as far north as Thunder Bay, over 800 km away. In this study, we focused on the months of May to September for their historically higher maximum daily temperatures in Ontario [18]. This also allowed us to exclude the effects of colder temperature on morbidity [19].

### 2.2. Population and Data Sources

We included the medical records of children and adults who presented to the EDs at LHSC between May and September 2014–2018, where at least one ICD-10 code for any diagnosis had been assigned to their medical record. Individual patient medical records were identified using the National Ambulatory Care Reporting System (NACRS) database. Once records were identified, we then extracted data from the electronic medical record at LHSC called Cerner (Cerner Corporation, Kansas City, MO, USA), which contains all the clinical information from the ED visit. In the case that paper charts were accessible, we also abstracted data from these files. Moreover, we included the daily temperatures at the time of the ED visits from Environment and Climate Change Canada’s London airport (ID: London CS) weather station data [20].

### 2.3. Administrative Coding Algorithm

We developed an algorithm of ICD-10 codes for AHI based upon a literature review and clinical expertise [13,14,15]. Our algorithm included codes for both heat-related illness (e.g., heat illness) and the consequences of heat-related illness (e.g., acute kidney injury), which we termed “expanded AHI codes”. These were also selected based on the previous literature and multidisciplinary clinical expertise (Table 1). We excluded codes related to “human-made” heat exposure codes as previously recommended (i.e., occupational hazards such as cooking and saunas—code W92) [7].

### 2.4. “Gold-Standard” Definition of AHI

Our “gold-standard” definition of AHI was defined as both healthcare provider documentation of patient-reported exposure to heat as well as documentation of at least one heat-related complaint (e.g., syncope, heat stroke, loss of consciousness, muscle cramps, pain, exhaustion, fatigue, edema, etc.). We validated this “gold-standard” definition with an unweighted kappa analysis.

### 2.5. Data Abstraction

Two independent reviewers reviewed and abstracted data from all medical charts. To establish the positive predictive value (PPV) of our administrative algorithm, we reviewed 62 randomly selected algorithm-positive charts (i.e., the medical chart was assigned at least one of our algorithm’s codes in any diagnostic field). To calculate the sensitivity (Sn), negative predictive value (NPV), and specificity (Sp) of our algorithm, we then reviewed a random sample of 964 ED charts that included any ICD-10 code. Blinded chart abstraction was completed in duplicate for 20% of the sample.

### 2.6. Outcomes

Our primary outcome was to establish the validity of our coding algorithm for AHI (i.e., PPV, NPV, Sn, and Sp). As additional outcomes, we hoped to examine the PPV of individual ICD-10 codes and the validity of our coding algorithm on hot days (defined as those where the maximum daily temperature reached ≥31 °C, which is the current threshold for heat warnings in Ontario [21], moderately hot days (those where maximum daily temperatures were 20–30 °C), and cooler days (those where maximum daily temperatures were <20 °C).

### 2.7. Statistical Analysis

For a precision level of 0.10 and a confidence level of 0.95, we estimated requiring a total of 62 records to be able to establish the PPV of the algorithm (based upon the estimated prevalence of AHI in Ontario) and 931 records to establish the NPV and Sn (Table 2).

## 3. Results

There were a total of 326,702 eligible ED encounters between May and September 2014–2018 at LHSC. Mean temperatures and the proportion of cool, moderate, and hot days over the study period are presented in Table 3.

### 3.1. PPV

Of the eligible ED encounters, 208 (0.06%) were assigned at least one of the ICD-10 codes in our AHI algorithm. We randomly selected 62 of these charts for detailed review to establish our PPV (Figure 1). The mean (SD) age of patients at the time of their ED encounter was 38.8 (23.8) years. Thirty-seven percent were female (Table 4). Seventy-nine percent of patients were discharged home following their encounter, while 4.8% were hospitalized. The overall positive predictive value (PPV) was 61.3 ± 12.1% (95% CI).

### 3.2. NPV, Sn, and Sp

We then randomly selected and reviewed an additional 964 ED encounters with any ICD-10 code to ascertain the NPV, Sn, and Sp of our algorithm (Figure 2). The mean (SD) age of patients at the time of their encounter was 41.7 (26.5) years, and 51.1% were female. (Table 4) Seventy-six percent of patients were discharged and 17.3% were hospitalized after their ED visit. Of these encounters, 960 records did not meet our “gold-standard” definition of AHI and were appropriately coded as a non-AHI encounter based on ICD-10 codes (i.e., negative AHI).

There were four records that met our “gold-standard” definition of AHI (i.e., “true AHI encounters”) upon detailed review. These encounters took place in 2016 and in 2018 on moderate and hot days (Table 3). Only one of these “gold-standard” AHIs was correctly assigned one of the ICD-10 codes in our algorithm. Thus, the overall Sn was 25.0 ± 42.4%, while the Sp was 100.0 ± 0.0% and the NPV was 99.7 ± 0.4% (Table 5). Confidence intervals were calculated using a standard formula according to the efficient-score method, corrected for continuity [23]. Given the small number of “gold-standard”-positive or “true” AHI encounters, we were unable to examine the validity of our algorithm on cool, moderate, and hot days, nor were we able to examine individual AHI ICD-10 codes.

We measured our inter-rater reliability by calculating an unweighted kappa using the 20% of overlapping reviews. Our unweighted kappa was 0.9 ± 0.1, representing excellent agreement.

## 4. Discussion

### 4.1. Main Findings

In this retrospective cohort study conducted over five years in a large academic setting in Ontario, Canada, we found that administrative codes for AHI were highly specific; in fact, the Sp of the algorithm was 100%. This suggests that algorithm-positive AHIs were true AHIs and that our algorithm can be used to confidently identify patients with heat-related illness. This is important for researchers who might be interested in capturing and following cohorts of patients who have experienced AHI in our region.

However, the algorithm lacked Sn, and with it, we were only able to capture one in four “true” AHIs. The reasons for these findings may be manifold. For one, we had a “strict” “gold-standard” definition of AHI. In addition to suggestive symptoms (e.g., syncope), we required that the patient-reported heat exposure be documented in the medical record by the treating physician or nurse. If care providers did not ask patients about heat exposure, if the patients did not voice this, or if care providers did not explicitly record exposure in the medical record, “true” heat illness would not have been captured. It is also possible that the low sensitivity of our algorithm resulted from the other co-diagnoses that patients presented to the ED with. For example, if patients also had acute shortness of breath requiring immediate medical attention, care providers may have been diverted away from taking a fulsome history and may not have captured the signs and symptoms of AHI or preceding heat exposure. Understanding whether the sensitivity of the algorithm improved with increasing temperature would have been extremely interesting but unfortunately not possible given the limited number of events. We did however observe that all encounters that met our “gold standard” occurred on moderate or hot days.

### 4.2. Results in Relation to Other Studies

There are few studies that examine the validity of administrative data for AHI, and to our knowledge, none specifically validate ICD-10 codes. A health record review by DeGroot et al. attempted to determine the validity of ICD-9 codes for AHI in a cohort of 290 heat illness casualties that occurred in U.S. Army soldiers from 2009 to 2012 [14]. In their study, “2 out of 3 of the following must have been met to be considered heat injury or heat stroke: aspartate transaminase/alanine transaminase fold increase >3, creatine kinase fold increase >5, and/or creatinine ≥ 1.5 mg/dL.” Based on this definition, there were 80 cases that met their diagnostic criteria, but of those, only 28 were diagnosed as such, providing a limited Sn of 65%. A total of 66 of 210 cases were incorrectly diagnosed as heat injury or heat stroke despite not meeting diagnostic criteria (Sp = 69%). PPV and NPV were 0.44 and 0.84, respectively. The Sn in their study was much higher than the 25% found in our study, which may be because we used a strict “gold-standard” definition of AHI (i.e., we required documented exposure to heat alongside a heat-related complaint). Moreover, our study was conducted in Canada, which can be more than 10 degrees Celsius cooler than the United States, where most other validation studies have been completed [24]. In contrast, we had a higher Sp, PPV, and NPV in our study. Overall, the authors suggest that caution should be exercised when examining epidemiological surveillance data on heat illnesses, as there was disagreement between their gold-standard diagnostic criteria and the selected ICD-9 code in approximately 1/3 (94/290) of all cases in their cohort. Based on our study’s results, we agree with their recommendation.

Another study by Harduar Morano and Waller analyzed ED visits in North Carolina between 2012 and 2014 for the purposes of improving AHI surveillance [13]. They evaluated and refined a heat syndrome case definition using keywords in triage notes or ICD-9 codes with additional heat-related keywords, then calculated the Sn and PPV of keyword-identified ED visits, and manually reviewed ED visits to identify true positives and false positives. Their initial definition identified 8928 ED visits, but after applying the revised inclusion and exclusion criteria (e.g., including keywords such as “heat ex”, “overheat”, “too hot, and “heat stroke”, while excluding keywords such as “burn”, “grease”, “liquid”, “oil”, “radiator”, “antifreeze”, “hot tub”, “hot spring”, and “sauna”), this increased to 9132 heat-related ED visits. Of those 9132 ED visits, 2157 (23.6%) were identified by a heat syndrome keyword only, 5493 (60.2%) by a heat-related diagnostic code only, and 1482 (16.2%) by both methods. Based on the additional keywords and exclusion criteria, both the Sn (1482 of 6975, 21.2%) and the PPV (1482 of 3639, 40.7%) improved from 15.9% and 36.7%, respectively. The Sn in this study was comparable to ours, though our PPV is higher. We agree with the authors’ conclusion that refinement of case definitions using inclusion and exclusion criteria beyond ICD codes are useful to improve the accuracy of surveillance systems for capturing the frequency of AHI.

It is also important to highlight that despite an overall lack of validation studies of ICD-10 codes for AHI, multiple previous studies have already used them to surveil AHI in various populations within the U.S. One study by Dring et al. used ICD-9 and ICD-10 codes and estimated that there were 1,078,432 ED visits for heat-related emergency medical conditions recorded nationally from 2008 to 2020 [25]. Furthermore, public health departments are also using ICD codes to monitor for AHI within their local jurisdictions. For example, a study by the Michigan Department of Health and Human Services, Division of Environmental Health, published online data with the intent to provide public health professionals, researchers, and the general public with summary information on heat illness ED visits in the State of Michigan [26]. Their dataset included ED visits with the same ICD-10 codes for heat illness as our study and similarly excluded heat exposure of human-made origin. Finally, a recently published study examined heat exhaustion and heat stroke among active members of the U.S. Armed Forces from 2019 to 2023 [27]. In this study, a case of heat illness was defined as an individual with (1) a hospitalization or outpatient medical encounter record with a primary or secondary diagnosis of heat exhaustion or heat stroke (according to corresponding ICD-10 codes) or (2) an internal Reportable Medical Events record of heat exhaustion or heat stroke. Using this definition, heat illness rates were higher among those younger than age 20, Marine Corps and Army service members, non-Hispanic Black members, and newer recruits. For all of these studies, conclusions need to be interpreted with caution as there may be important disagreement between the frequency of AHI reported based on ICD codes versus gold-standard AHI definitions. Our study’s coding algorithm, which improves upon strictly using ICD-10 codes, can thus be incorporated in future research methods when investigating and monitoring for AHI using administrative databases in other populations.

Although our study was conducted in one of the most southern parts of Canada, our geographical area still has a temperate climate compared to other jurisdictions. In warmer or tropical regions where there may be a greater severity and longer duration of environmental heat, ED patients with AHI may present more frequently and with more severe symptoms and related morbidity (that might be more likely to be documented). For example, a study by Harduar Morano and Watkins used multiple data sources to improve the sensitivity of AHI surveillance in their study, including ED visit records, hospital discharge records, and death certificates [15]. They described that a total of 23,981 ED visits, 4816 hospitalizations, and 140 deaths in Florida were due to heat illness and about 20.1% were hospitalized from 1 May to 31 October, 2005–2012, compared to only 4.8% in our sample of patients. As climate change is expected to increase the frequency of extreme heat events and raise ambient temperatures, we expect that trends of ED visits, hospitalizations, and deaths will only increase, even in temperate climates. The aforementioned study by Dring et al. demonstrated that the annual incidence rate of ED visits per 100,000 population increased by an average of 2.85% per year, ranging from 18.21 in 2009 to 32.34 in 2018, with a greater total visit burden greatest in the Southern U.S. [25]. However, the authors noted that there was an overall increase in heat-related ED visits affecting patients in all regions and during all seasons.

### 4.3. Strengths and Limitations

Our “gold-standard” definition differentiates this study from previous analyses, as it was based on an extensive literature review and clinical consultant expertise. We had a high inter-rater reliability, as suggested by our unweighted kappa. Where most validation studies focus solely on PPV, we also provide NPV, Sn, and Sp.

Our study is limited by clinical documentation, which is a limitation of all health records reviews. This issue can be particularly apparent in the ED; in prior studies, there has been a noted reduction in the quality/comprehensiveness of medical documentation in the ED setting relative to other care settings [15]. The use of ICD codes is also associated with limitations. ICD codes are designed for billing rather than surveillance purposes, and assignment practices carry an inherent subjectivity and regional variability. We attempted to capture this variability by including two large tertiary care hospitals with diverse patient characteristics, but both will reflect the assignment practices of London, Ontario.

We decided to exclude ICD-10 code W92 (“man-made” heat) to increase Sp for environmentally caused AHI. This exclusion may have caused the exclusion of true AHIs, as there may have been miscoding. Moreover, we observed that with a true heat illness, patients often had physical exertion preceding their encounter. Including a code for physical exertion in our algorithm may have improved its Sn, but unfortunately, there is not one available. Additionally, our study focuses on the validity of ICD-10 codes for AHI as a collective whole, and our algorithm does not differentiate between individual codes or specific diagnoses (e.g., heat stroke versus heat syncope, etc.).

Lastly, a recent study by Sorsensen and Hess elucidated that the risk of heat-related illness is driven by heat exposure, individual susceptibility (age, pregnancy status, and coexisting conditions), and sociocultural factors (including environmental racism, poverty, lack of social cohesion, lack of access to healthcare, and limited worker protections) [28]. Our study unfortunately did not allow for the influence of socioeconomic characteristics on ED encounters for AHI.

## 5. Conclusions

Our ICD-10 coding algorithm for AHI had a high Sp and NPV but is limited in Sn. This algorithm can be used to assemble and study cohorts of patients who have AHI encounters. Its use, however, is limited in studying the true incidence and prevalence of AHI events. In order to actively capture AHI events, monitoring systems with a focus on prospectively detecting AHIs as they present to the ED might be helpful.

## Figures and Tables

**Figure 1 ijerph-21-01132-f001:**
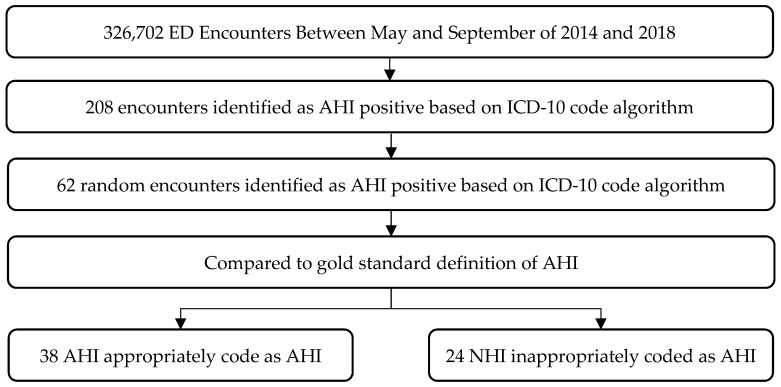
Review of 62 random algorithm-positive encounters. ED: Emergency Department, AHI: Acute Heat Illness, NHI: Non-Heat Illness.

**Figure 2 ijerph-21-01132-f002:**
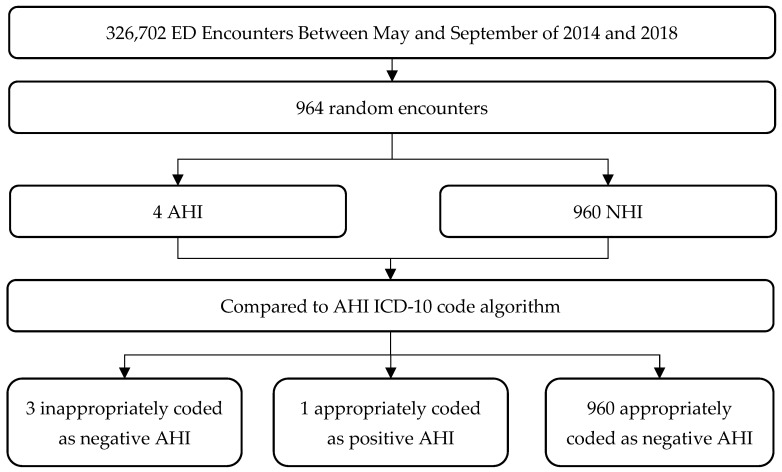
Review of 964 random ED encounters. ED: Emergency Department, AHI: Acute Heat Illness, NHI: Non-Heat Illness.

**Table 1 ijerph-21-01132-t001:** ICD-10 codes used in our algorithm for identifying acute heat illness in the emergency department. An asterisk represents a code that was not included. Expanded acute heat illness codes allowed for capture of complications, based on past literature and multidisciplinary clinical expertise.

ICD-10 Code	Description
X30	Exposure to excessive natural heat
W92 *	Excessive heat due to human-made conditions
X32	Exposure to sunlight
T670	Heat stroke and sunstroke
T671	Heat syncope
T672	Heat cramp
T673	Heat exhaustion, anhidrotic
T674	Heat exhaustion due to salt depletion
T675	Heat exhaustion, unspecified
T676	Heat fatigue, transient
T677	Heat edema
T678	Other effects of heat and light
T679	Effect of heat and light, unspecified
E860, E868	Volume depletion
N17	Acute kidney injury
R508	Other specified fever (hyperthermia)
R509	Fever, unspecified
M628, T296, G210	Rhabdomyolysis
E87	Sodium disorders
E875	Potassium disorders, hyperkalemia
R55	Syncope and collapse

**Table 2 ijerph-21-01132-t002:** Sample size values.

Sample Size Calculation for Positive Predictive Value	Sample Size Calculation for Sensitivity and Specificity
Estimated positive predictive value = 0.80 Desired precision = 0.10 Confidence level = 0.95 Total sample size = 62	Estimated sensitivity = 0.40 Estimated specificity = 0.90 Expected prevalence = 0.10 Desired precision = 0.10 Confidence level = 0.95 Total sample size = 931 *

Sample size formulae in Hajian-Tilaki K. Sample size estimation in diagnostic test studies for biomedical informatics. Journal of Biomedical Informatics 2014; 48: 193–204 [22]. * Discrepancy between sample size calculation (931) and number of reviewed health records (962) is due to available resources allowing for the revision and inclusion of 31 additional random records.

**Table 3 ijerph-21-01132-t003:** Median maximum temperature from a single weather station (at London Ontario International Airport) between May and September of 2014–2018 and proportion of cool, moderate, and hot days. Cool, moderate, and hot days were defined as days with a maximum temperature below the first quartile, between the first and third quartile, and above the third quartile of median temperatures for that year.

Year	Maximum Temperature (°C)		Proportion of Days by Temperature		
	Median	IQR	Cool	Moderate	Hot
2014	24.2	6.1	36%	45%	19%
2015	24.5	4.5	19%	60%	20%
2016	26.1	5.7	20%	47% *	33% *
2017	24.3	6.6	28%	47%	25%
2018	25.7	6.0	21%	43% *	36% *

* True AHI encounters (i.e., those that met our “gold-standard” definition).

**Table 4 ijerph-21-01132-t004:** Characteristics of 62 randomly selected algorithm-positive charts and 964 randomly selected non-algorithm-positive medical charts.

	Algorithm-Positive Medical Charts (n = 62)	Non-Algorithm-Positive Medical Charts (n = 964)
Mean age at time of encounter (SD)	38.8 (23.8)	41.7 (26.5)
Age range	0–92	0–102
65 years and above	5 (8.1)	223 (23.1)
Male	39 (62.9)	469 (48.7)
Female	23 (37.1)	493 (51.1)
Canadian Triage and Acuity Scale		
Resuscitation	2 (3.2)	38 (3.9)
Emergent	19 (30.6)	241 (25.0)
Urgent	31 (50.0)	444 (46.1)
Less Urgent	9 (14.5)	224 (23.2)
Non-urgent	1 (1.6)	11 (1.1)
Not recorded	0 (0)	6 (0.6)
Presenting Complaint		
	Heat Stroke 9 (14.5)	Abdominal pain 87 (9.0)
	Heat Exhaustion 7 (11.3)	Chest pain 53 (5.5)
	Headache 6 (9.7)	Lower extremity pain 30 (320)
Disposition		
Discharged home	49 (79.0)	729 (75.8)
Admitted to hospital	3 (4.8)	167 (17.3)
Left against medical advice	10 (16.1)	64 (6.6)

Note: Data are presented as numbers (percent total) unless otherwise indicated.

**Table 5 ijerph-21-01132-t005:** Calculation of sensitivity, specificity, and negative predictive value. AHI: Acute Heat Illness, TP: True Positive, FP: False Positive, FN: False Negative, TN: True Negative, Sn: Sensitivity, Sp: Specificity, NPV: Negative Predictive Value.

	“True AHI Encounters”	Not “True AHI Encounters”	
ICD-10 code-positive	TP = 1	FP = 0	
ICD-10 code-negative	FN = 3	TN = 960	NPV = 960/(960 + 3) = 99.7%
	Sn = 1/(3 + 1) = 25%	Sp = 960/(960 + 0) = 100%	

## Data Availability

The raw data supporting the conclusions of this article will be made available by the authors on request.

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
