# Peer review of "Validity of an ICD-10 Coding Algorithm for Acute Heat Illness in the Emergency Department: A Retrospective Cohort Study"

_ijerph, 2024, doi:10.3390/ijerph21091132_

Round 1

Reviewer 1 Report

Comments and Suggestions for Authors

-          Introduction: add a sentence clarifying why having validated ICD-10 codes is important

-          Methods

o   Lines 74-83: please clarify if NACRS and Cerner records were linked on the same patient? Or is that NACRS and Cerner were both different sources of individual AHI cases? It’s not very clear. Same with the paper chart- these were pulled for a patient already identified via NACRS? This section just needs a little bit more information

o   Why capture lab records (Cerner)  if not part of your AHI definition?

o   Section 2.4: assuming there was a pre-determined list of heat-related complaints? Please clarify

o   Section 2.4: please consider changing the wording “gold-standard” definition throughout the manuscript. Instead, this is just the operational definition used for the study. Gold standard is a common term used in healthcare professions to represent what should be in daily clinical practice. The definition used in this study is fine for the research design and purpose, but is definitely not what a clinician should be using in daily practice to diagnosis a heat illness.

-          Table 4: consider rewording the column titles. The 62 algorithm positive charts were also randomly chosen, as well as the 964 charts. This is confusing that the two columns are representing different portions of the study.

-          Discussion

o   Line 178 does not match with section 2.4. healthcare provider document exposure to heat is not mentioned in the methods, it says patient reported. Can this be clarified in the methods please

o   Line 215: how do you know this? This variable is not mentioned in the results? Or was this casually noted in chart review? Please clarify

o   Section 4.2: expand this section to include a couple of other ED AHI studies so there is a more robust compare/contrast of the existing literature as compared to this study.

o   Somewhere in the discussion section make mention that this study was focused on AHI as a collective whole. That the algorithm did not capture if individual diagnoses (heat syncope vs heat stroke) were accurately applied.

Reviewer 2 Report

Comments and Suggestions for Authors

This study has drawn on a large data base of ED encounters at two tertiary hospitals from May to September for the period 2014-2018. 

The study examined the relation between two parameters, an algorithmically-defined index of acute heat illness (AHI), based on ICD-10 coding assigned to ED patients, and a gold-standard for AHI, based on a patient-reported heat exposure together with documentation of a heat-related complaint.

Daily temperatures were obtained  for London Ontario – presumably these were daily maximum temperatures - but it is unclear how this information was used.  The definition of gold standard AHI included manifestation of AHI on a hot day, which the authors have defined as ≥310C, but according to Table 3, some cases of true AHI occurred on days of only moderate heat. There needs to be some explanation as to how the anomaly was reconciled.

Some explanation is need on how consequences of AHI, such as acute kidney injury, were included in the algorithm since they are not specified in the codes listed in Table 1.

Another concern is the relevance of the readings in London, since the catchment of the hospitals extended as far as 800km from the city, where maximum temperatures could be quite different. It would be preferable to report the maximum temperature at the weather station closest to where the AHI was experienced.

The aim of the study was to measure the validity of the coding algorithm for AHI. The measures of validity were sensitivity (Sn, specificity (SP), positive predictive value (PPV,) and negative predictive value (NPV).    These are measures of the relationship between a given test, which in this case is the algorithm, and a reference parameter which is presumed to be the true occurrence of the disease, represented in this case by the gold standard.

The first part of the study was to measure the PPV using a sample of 62 test-positive subjects.  This estimate is sound, although an explanation is needed on how the patient-recorded complaint on a hot day was interpreted when in fact the ambient temperature was <310C.

What happened in the second part of the study is less clear.   From Figure 2 it appears that the random sample of 964 patients was submitted to two successive screenings.  The second is stated to have been comparison with the AHI iCD-10 coded algorithm.   I therefore assume (although it is not stated) that the first step was to examine the 964 records to identify true disease cases, ie AHI defined by the gold standard. Of these 4 are identified as AHI. It is therefore to be expected that this would be reported as 4 cases of true AHI, all  of which were also test-positive, (ie were included in the algorithm). Instead, in lines 155-6 it is expressed the other way around, ie that of the 4 cases who were test-positive, all 4 were true cases.

In the next step on the flow chart, all 964 cases were screened for test-positivity (the test being the algorithm). Lines 156-7 state:

“There were four records that also met our gold standard definition of AHI but did not contain any of the ICD-10 codes in our algorithm.”

Since the four true cases previously identified were test-positive, this presumably means that these were four more cases, so that there are now 8 true cases.  But why were these 4 true cases not detected in the first screening?

One would have expected that, having identified 4 true cases in the first screen, the remaining 960 cases would be screened for test-positivity, thereby getting an estimate of specificity. It is not clear why all 964 cases were included.

The text then states (lines 159-60):

“There were four 156 records that also met our gold standard definition of AHI but did not contain any of the 157 ICD-10 codes in our algorithm.”

This appears (at least to this reviewer) as a contradiction of the previous line quoted.

I am unable to reconcile the text with Figure 2, and am therefore unable to understand how the indexes of validity (other than the PPV in the first part) were derived.

Round 2

Reviewer 2 Report

Comments and Suggestions for Authors

The authors have clarified their estimates of sensitivity, specificity and NPV in the authors’ response.  The text has been revised so that the flow diagram and the estimates are now clear to the reader.  I suggest you consider including the table that you included in the authors’ response. There should also be some indication of how the confidence intervals were computed.

There is still some ambiguity around the inclusion of days in which the maximum temperature was less than the threshold level for heart warnings in Ontario. The gold standard criteria include not only patient-reported exposure to heat (line 103), but also that the condition occurred while working on a hot day (line 105). While it is of course true that heat-related illness can occur on days when the air temperature does not exceed 310C, your gold standard precludes occurrences on such days.  The footnote to Table 3 stating that true AHI occurred on days in some of which heat levels were only moderate, is not correct.
